# Mucosal Antibodies: Defending Epithelial Barriers against HIV-1 Invasion

**DOI:** 10.3390/vaccines7040194

**Published:** 2019-11-23

**Authors:** Ruth M. Ruprecht, Bishal Marasini, Rajesh Thippeshappa

**Affiliations:** 1New Iberia Research Center, University of Louisiana at Lafayette, New Iberia, LA 70560, USA; bishal.marasini@louisiana.edu; 2Texas Biomedical Research Institute, San Antonio, TX 78227, USA; rthippeshappa@txbiomed.org

**Keywords:** mucosal HIV-1 transmission, mucosal antibodies, recombinant monoclonal IgM, IgG and dimeric IgA, passive mucosal immunization, SHIV, mucosal barriers, immune exclusion

## Abstract

The power of mucosal anti-HIV-1 envelope immunoglobulins (Igs) to block virus transmission is underappreciated. We used passive immunization, a classical tool to unequivocally prove whether antibodies are protective. We mucosally instilled recombinant neutralizing monoclonal antibodies (nmAbs) of different Ig classes in rhesus macaques (RMs) followed by mucosal simian–human immunodeficiency virus (SHIV) challenge. We gave anti-HIV-1 IgM, IgG, and dimeric IgA (dIgA) versions of the same human nmAb, HGN194 that targets the conserved V3 loop crown. Surprisingly, dIgA1 with its wide-open, flat hinge protected 83% of the RMs against intrarectal R5-tropic SHIV-1157ipEL-p challenge, whereas dIgA2, with its narrow hinge, only protected 17% of the animals—despite identical epitope specificities and in vitro neutralization curves of the two dIgA isotypes (Watkins et al., *AIDS* 2013 27(9):F13-20). These data imply that factors in addition to neutralization determine in vivo protection. We propose that this underlying protective mechanism is immune exclusion, which involves large nmAb/virion aggregates that prevent virus penetration of mucosal barriers. Future studies need to find biomarkers that predict effective immune exclusion in vivo. Vaccine development strategies against HIV-1 and/or other mucosally transmissible pathogens should include induction of strong mucosal Abs of different Ig classes to defend epithelial barriers against pathogen invasion.

## 1. Introduction

Traditionally, vaccine development against viruses has relied on inducing antibody (Ab) responses, which in most cases, are the readout for “vaccine take”. The initial strategy to develop vaccines against HIV-1/AIDS was explored with the use of monomeric gp120 as immunogen. Repeated Phase III trials showed no efficacy [1,2]. Since then, the key goal has been to generate newer immunogens that can induce broadly reactive neutralizing antibodies (bnAbs). Thus far, this approach has not succeeded.

For the sake of clarity, it is important to consider the narrow definition of neutralization in the HIV-1 field. It is based upon standardized in vitro assays using cell lines. As such, the term “neutralization” defines the ability of an Ab to block virus entry into a target cell, with the readout being activation of a reporter gene or release of infectious virus into tissue culture supernatants. The susceptibility of various strains of HIV-1 to neutralizing Abs (nAbs) in vitro is described using a tier system, according to which standard panels of neutralizing monoclonal antibodies (nmAbs) or well-defined sera collected from HIV-1-infected individuals harboring different clades of the virus are used as references. As such, tier 1 HIV-1 isolates are easy to neutralize and typically represent cell line-adapted strains. Tier 2 viruses are more difficult to neutralize but are not resistant across-the-board to the test reagents. Most individuals with newly diagnosed HIV-1 infections harbor tier 2 strains that only use CCR5 as a coreceptor for target cell entry. Finally, tier 3 viruses are resistant to neutralization by the nmAbs or sera on the standard panels. These in vitro analyses are helpful to allow cross-comparison of nAb responses induced by investigational vaccine regimens. However, the assay readout is only focused on blocking entry of virus into a target cell in vitro.

To develop vaccines against HIV-1/AIDS, it is important to keep in mind that 90% of all acute infections occur through mucosal exposures, which includes sexual as well as perinatal transmission of HIV-1. Most vaccine approaches under development have predominantly focused on analyzing Ab- or cell-mediated immune responses in the blood compartment. However, Abs can greatly impact virus acquisition at mucosal levels. A number of groups have performed passive immunizations in nonhuman primates using *mucosal* application of nmAbs and shown protection against simian–human immunodeficiency virus (SHIV) challenge [3,4,5,6,7,8,9,10]; reviewed in [11] (summarized in Figure 1B).

We have proposed that mucosally delivered nmAbs keep the virus at bay through a potent mechanism—immune exclusion (reviewed in [15,16]). Antiviral Abs produced locally at the mucosal barriers are transported across them by special receptors into the luminal space where the Abs can trap incoming virus. Preventing the overwhelming majority of infectious virions from crossing the epithelial barrier will have a major impact on preventing systemic virus spread. Thus, retaining infectious HIV-1 particles at mucosal linings is the first major defense mechanism that prevents the virions from getting close to their target cells. Using passive immunization as an investigational tool that can clearly delineate cause and effect, we have shown that anti-HIV-1 dimeric IgA (dIgA), IgG [6], and—most recently—also pentameric IgM [10] are effective defenders at the mucosal level.

Preventing most infectious virus particles present in an inoculum from crossing the mucosal barriers will decrease the attack force that may penetrate and establish a beachhead beyond the front line. Anti-HIV-1 Abs present in mucosal tissues or in the systemic circulation thus will have to contend with much fewer virions, making it possible to stop further virus spread or slowing it considerably, resulting in delayed peak viremia and lower steady-state viral loads. In such a scenario, Ab effector functions can have a major impact on the overall outcome. The ability of Abs to kill infected cells in the context of Ab-dependent cellular cytotoxicity (ADCC) and other mechanisms can theoretically be effective in the absence of neutralization in the strict sense of the definition.

Lastly, Abs can also trigger priming of cell-mediated immune responses. This implies that Ab-virion complexes are taken up and delivered to antigen-presenting cells (APCs) through receptor-mediated interactions with immune complexes [17]. 

The goals of this review are to: (i) discuss the role of different classes of immunoglobulins (Igs), IgM, IgG, and IgA, as mucosal defenders; (ii) consider local interactions of mucosal Igs with each other, mucus, and the epithelial tissue environment; and (iii) address unanswered questions regarding the mechanisms of immune exclusion.

Of note, the major focus of our review is on *mucosal* Abs—not on the prevention of mucosal SHIV transmission in general terms. Most passive immunization studies performed since the late 1990s involved systemic administration of monoclonal or polyclonal Abs, using either intravenous, intramuscular, subcutaneous or intraperitoneal routes. Such studies are beyond the scope of our review that seeks to emphasize the role of *mucosal* Abs. Thus, we summarize nonhuman primate studies, in which the Abs were administered through one of the *mucosal* routes (for passive immunizations). For active immunizations, we have reported the few studies in which mucosal Abs were analyzed in detail and found to impact the outcome. 

## 2. Generation and Transepithelial Transport of Mucosal Igs

### 2.1. Secretory IgM (SIgM)

IgM can exist either as pentamer or hexamer, with the latter lacking the joining (J) chain [18,19]. The J chain plays a critical role in transporting IgM across the mucosal barrier [20,21]. IgM found in mucosal fluids is produced by subepithelial plasma cells that add the J chain to pentameric IgM. This form of IgM interacts via the J chain with the polymeric immunoglobulin receptor (pIgR), which is expressed on the basolateral surface of epithelial cells. Complexes of pIgR-IgM are taken up by transcytotic vesicles and shuttled across the epithelial cell towards the luminal side. There, pIgR is cleaved, leaving behind a cell-membrane-bound stump and the secretory component (SC) that stays with the IgM molecule, thereby forming secretory IgM (SIgM) (Figure 1A, top section). 

The stoichiometry of binding between pentameric, J chain-containing IgM with SC is believed to be equimolar [22]. All mucosal fluids tested in different species are known to also contain free SC. The latter is believed to confer some degree of protection to Igs against cleavage by the various proteases present in the mucosal fluid environment.

### 2.2. Mucosal IgG: Two Sources

Unlike SIgM, IgG found in mucosal fluids can have two different sources: (i) the systemic compartment, and (ii) local production by IgG-secreting plasma cells in the lamina propria. IgG molecules are known to extravasate from blood vessels, permeate tissues, and are present in mucosal fluids. The production of systemic IgG occurs in the bone marrow from where it is taken up in the systemic circulation. IgG produced by local subepithelial plasma cells or present in mucosal tissues after extravasation binds to the neonatal Fc receptor (FcRn) (Figure 1A, middle section). FcRn shuttles back and forth between the basolateral and luminal sides and does not undergo proteolysis—unlike pIgR. Once IgG is loaded onto FcRn in the subepithelial space, the complex is transported inside transcytotic vesicles and IgG is released into the mucosal fluid at the luminal side, a process that is pH-dependent. Unlike Igs transported by pIgR that gets modified by the addition of SC, IgG released from FcRn into mucosal fluids is not altered and remains monomeric. FcRn is known to bind to immune complexes (ICs) formed by mucosal fluid IgG and foreign antigens and to transport such ICs back across the epithelial barrier. 

### 2.3. Secretory IgA (SIgA)

Among all mucosal Ab isotypes, IgA is remarkable as it is the most abundant Ab. The human body produces more IgA per day than all other Ig classes combined. A recent review [23] reported a synthesis rate of 66 mg/kg each day. Of note, the plasma concentration of IgA in humans is lower than that of IgG, indicating that most of the IgA does not end up in the systemic circulation. Instead, most IgA is produced locally by subepithelial plasma cells and released into mucosal fluids that need to be replaced continually. Clearly, the role of IgA in the mucosal environment is of critical importance—given that the body invests such resources for the constant steady replenishment of IgA. 

The individual steps of mucosal IgA production are depicted in Figure 1A (bottom section). Plasma cells in the lamina propria release IgA in the form of dIgA that contains the J chain. As described above for the generation of SIgM, dIgA binds to pIgR expressed on the basolateral surface of epithelial cells. After the pIgR-dIgA complex is formed, it enters a transcytotic vesicle that traverses the epithelial cell. Upon reaching the luminal side, pIgR undergoes proteolytic cleavage, leaving behind a pIgR remnant on the cell surface and the SC component that now forms secretory IgA (SIgA). Essentially, the pIgR-based mechanism is used to deliver pentameric IgM and dIgA into mucosal fluids such as SIgM and SIgA, respectively. 

The ratios of mucosal IgM:IgG and IgM:IgA can vary considerably because the generation of mucosal IgG and IgA depends on CD4^+^ T-helper cells to provide critical stimulatory signals to B cells for Ig class switching. Different species have different IgA subtypes. Humans and the great apes have two isotypes, IgA1 and IgA2, which show great structural differences in the Ig hinge region. IgA1 has a wide-open long hinge region that is 19 amino acids (AA) long and contains a number of O-linked oligosaccharides. In contrast, the human IgA2 hinge region is short (only six AA) and lacks glycosylation. IgA1 molecules with their wide-open hinge region assume a T-like shape where the distance between Fab fragments is ~16 nm. This contrasts with the IgA2 hinge that has the typical Ig Y shape, with the distance between Fab regions of only 10 nm. While the IgA1 hinge is flexible, that of IgA2 is short and stiff. Little is known about the mechanisms that control isotype switching to IgA1 versus IgA2. Because only humans and great apes have the IgA1 subclass, no practical animal models are available to examine IgA isotype switching. Rhesus macaques (RMs) only harbor the IgA2-like subtype (reviewed in [15]). 

T-helper-cell-independent mucosal IgA responses do occur as well [12,13]; such Ab responses seem to predominantly focus on commensal bacteria, rather than viral pathogens such as HIV-1. The latter pathogen causes marked skewing of the IgM:IgG and IgM:IgA ratios, which implies that most of the HIV-1-specific mucosal B-cell responses are dependent on T-helper cells. Even early in infection, these T-helper cells are seriously reduced in number, which leads to almost undetectable IgA responses in mucosal fluids of some HIV-1-infected individuals. 

### 2.4. Mucosal IgM, IgG, and IgA as Defenders of Mucosal Integrity

Indirect evidence points to the critical importance of all three classes of Ig at mucosal barriers. In congenital selective IgA deficiency, the majority of affected individuals are asymptomatic. In this condition, IgM combined with IgG can mostly compensate for the seriously decreased concentrations or lack of IgA in mucosal fluids. This implies a redundancy built into mucosal defenses such that IgM and IgG can compensate for the function(s) normally provided by IgA. These observations imply a cooperative interaction among the three classes of Igs in the mucosal environment. 

In HIV-1 and SIV infection, even in the early acute infection stage, CD4^+^ T-helper cells are severely affected and their numbers are drastically reduced due to virus-induced cell killing. This results in changes in the IgM:IgG and IgM:IgA ratios in mucosal fluids, given that IgM production does not depend on class switching and the necessary stimulation from functionally intact CD4^+^ T-helper cells. Mestecky et al. [24,25] first noticed markedly diminished anti-HIV-1 IgA responses in mucosal fluids. Severe immunodepletion of CD4^+^ T cells in gut lamina propria was first reported by Smit-McBride et al. [26]. The significant changes in the composition of mucosal fluid Igs results in the loss of epithelial barrier integrity and the “leaky gut syndrome”, the hallmark of which is increased plasma concentration of lipopolysaccharide (LPS) [27,28]. 

## 3. Passive Mucosal Immunization: Proof-of-principle that Anti-HIV-1 dIgA, IgG, and IgM Can Prevent Virus Acquisition after Mucosal Challenge

Until recently, the protective role of anti-HIV-1 IgA has been debated. Data from the RV144 Phase III vaccine efficacy trial [29] implied that systemic IgA directed against HIV-1 Env interfered with the protection provided by vaccine-induced IgG [30]. The latter mediated protection via its ADCC effector function, rather than neutralization [31]. This large RV144 study did not assess mucosal Ab responses. 

Our group has provided direct evidence that mucosal Igs can protect. We performed our studies in RMs that were exposed to R5-tropic SHIV strains carrying an HIV-1 clade C envelope known to be recently transmitted [32,33]. The data are summarized in Figure 1B [3,4,5,6,7,8,9,10]. In the first study [6], we performed passive mucosal immunization and tested a panel of recombinant human mAbs engineered to have the same epitope specificity conferred by the VH and VL genes in the context of human dIgA1, dIgA2, and IgG1 backbones (Figure 1B, middle and bottom sections). The VH and VL genes had been derived from the IgG1 mAb HGN194 isolated from an individual chronically infected with a circulating recombinant HIV-1 form of clades A/G (HIV_CRF A/G). HGN194 recognizes the conserved crown of the HIV-1 V3 loop and neutralized all R5 tier 1 strains tested, as well as several tier 2 strains [34]. 

In the first mucosal passive immunization study [6], RMs were given HGN194-dIgA1, HGN194-dIgA2, or HGN194-IgG1—all by the intrarectal route at the same total milligram dose. Control RMs were left untreated. Thirty minutes after the topical mAb administration, all RMs were challenged intrarectally with a high dose of R5-tropic SHIV-1157ipEL-p, a tier 1 strain that carries a recently transmitted pediatric HIV clade C envelope [33]. All controls became viremic, whereas 83% of dIgA1-treated RMs were protected. In contrast, only 17% of those given the dIgA2 version and 33% of those given the IgG1 isotype remained aviremic. When comparing the dIgA1-treated RMs with those given the dIgA2 isotype, the difference was statistically significant. Importantly, better in vivo protection was not a function of differences in neutralization profiles but correlated with a better ability of dIgA1 to capture virions and to block transcytosis in a transwell-based assay compared to the dIgA2 form [6]. This observation raises the possibility that protection may not be linked to the ability to neutralize, but rather to capture virus particles in the local mucosal environment.

When considering the presence of different Ig classes in mucosal fluids, we came to realize that no information existed for the potential of anti-HIV-1 IgM to block mucosal virus transmission. Therefore, we generated 33C6-IgM [10] and tested it with its IgG1 counterpart in the same RM/SHIV mucosal challenge model system (Figure 1B, top section). The recombinant form of 33C6-IgM was generated from mAb 33C6-IgG1 [35], which also recognizes the same conserved V3 loop crown in HIV-1 Env as HGN194.

We also used the same total milligram doses of recombinant IgM and IgG1 mAbs, followed 30 min later by a single high-dose R5 SHIV challenge by the intrarectal route. Again, all untreated controls became highly viremic. In contrast, 67% of RMs pretreated topically with the IgM form were protected and 83% of those given the IgG1 form. The degree of protection between IgM and IgG1 forms was not significantly different—even though five-times fewer IgM molecules had been given considering that the same milligram amounts were administered using the pentameric IgM versus the monomeric IgG1 forms. 

Taken together, our passive immunization studies demonstrate clearly that mucosally delivered pentameric IgM, dIgA1, as well as dIgA2 and IgG1 versions with the same epitope specificity, can provide significant protection against mucosal R5 SHIV transmission. Among the different isotypes, the dIgA2 version provided the lowest degree of protection, whereas dIgA1 and IgM protected a higher fraction of the passively immunized RMs. These studies established a novel paradigm: Anti-HIV-1 mAbs of different classes can protect within the mucosal compartment. As such, this is important information for designing active immunogens with the aim to induce Ig-mediated protection along the mucosal lining. 

We postulate that the mechanism of protection provided by the topical intrarectal administration of the various classes of mAbs was *immune exclusion*. According to this mechanism, mAbs trap incoming virus particles and form large networks to prevent virus transcytosis across the epithelial barrier. Not surprisingly, multimeric mAbs with their improved ability to crosslink virions performed well. A number of questions regarding immune exclusion remain to be addressed. In our proof-of-principle passive mucosal immunizations, we have targeted a protruding epitope, namely the exposed V3 loop crown that is highly conserved among tier 1 viruses. It is unknown whether mAbs targeting more recessed epitopes on HIV-1 Env will provide the same degree of protection. Importantly, the mAbs we used were neutralizing—although neutralization was not the correlate that explained the significant difference in protecting RMs against SHIV challenge between the dIgA1 and the dIgA2 forms of HGN194. It is possible that the Abs capable of immune exclusion need not to be neutralizing, as long as they can rapidly bind to incoming virions and ensnare them in large immune complexes. Another unanswered question is the relative potency of IgM versus IgG1 versions. As mentioned, all experiments summarized above used the same total mg amount, which corrects for valency but not molarity. It is conceivable that on a per molecule basis, IgM can provide superior protection.

## 4. Synergistic Interaction of Systemically and Mucosally Administered Igs in the Local Mucosal Environment

Prompted by the unexpected post-trial analyses of RV144, which implied that systemic anti-HIV-1 Env IgA antibodies interfered with the protective action of systemic IgG [30], we sought to examine the interaction between IgA and IgG versions of the same nmAb in the mucosal as opposed to the systemic compartment. The RV144 findings led us to postulate that the IgA version would also diminish the protective efficacy of its IgG counterpart in the mucosal environment—that IgA would act as the “bad guy” and block the beneficial effects of IgG. We, therefore, chose to perform our study with dIgA2 isotype of HGN194, which in our previous study, has only protected 17% of the RMs when used as a single agent against intrarectal SHIV challenge [6]. As mentioned above, the HGN194 dIgA1 isotype had protected 83% of the RMs and had been more potent than the intrarectally administered HGN194 IgG isotype (33% protection). Rather than administering IgG mucosally, we decided to treat the RMs with a suboptimal dose at the intravenous route 24 hours before the single, high-dose SHIV challenge. We did this to mimic IgG extravasation and distribution in tissues followed by transepithelial transport into the mucosal lumen. 

Our study [8] involved three groups of RMs; Group 1 received only intravenous treatment with HGN194 IgG 24 hours before SHIV challenge. Group 2 was treated with the same IgG regimen plus an additional intrarectal passive immunization with dIgA2 30 min before SHIV challenge. Control animals were left untreated. When given alone, low-dose intravenously administered HGN194-IgG1 provided no protection. The dIgA2 isotype given as single agent had previously protected 17% of intrarectally treated RMs. Unexpectedly, combining low-dose intravenously administered HGN194-IgG1 with topically administered dIgA2 protected 100% of the intrarectally challenged RMs [8]. This surprise finding could not be explained by synergistic in vitro neutralization as the nmAbs targeted the same epitope and thus competed with each other. We hypothesize that the mucosal fluid environment is fundamentally different from the conditions typically encountered by the virus in a standard vitro neutralization assay. Epithelial barriers are known to have different mucins that can interact with Igs. We have been able to replicate our earlier unexpected finding in a repeat study and again achieved 100% protection with suboptimal doses of intravenously administered IgG and intrarectally delivered dIgA2 version of HGN194 (Dr. Ruth Ruprecht et al., unpublished data). 

## 5. Induction of Adaptive T-cell Immunity by Mucosal Immune Complexes (ICs)

In 2011 [17], we first showed cross-clade protection by passive immunization with the anti-V3 loop nmAb HGN194-IgG1 [34]. RMs treated intravenously 24 hours before a single high-dose intrarectal SHIV challenge were completely protected at a dose of 50 mg/kg. The lower dose (1 mg/kg) showed partial protection; the RM with breakthrough infection had delayed peak viremia [17]. 

Next, we examined the completely protected RMs that never had any viremia for signs of host responses to the live-virus exposure under nmAb coverage. We postulated that the passively administered HGN194-IgG1 formed ICs with incoming virus particles and that such ICs would be taken up by antigen-presenting cells and possibly induce T-cell responses. To test this hypothesis, we performed interferon-γ (IFN-γ) ELISPOT assays using peripheral blood mononuclear cells (PBMC) from the protected animals stimulated ex vivo with overlapping synthetic peptides representing the open reading frames of SIV Gag, Nef, and HIV-1 Tat. In parallel, we also performed T-cell proliferative assays using PBMC stimulated with SIV Gag, HIV-1 Env, or HIV-1 Tat proteins. All nmAb-treated, protected RMs tested negative by all ELISPOT assays, whereas in three out of four virus-only control viremic RMs revealed positive ELISPOTs. Surprisingly, all of the protected RMs given passive immunization revealed proliferation of their CD4^+^, as well as their CD8^+^ T cells, following ex vivo stimulation with SIV Gag but after stimulation with HIV-1 Env or Tat proteins. These data suggest that specific memory T cells developed only against the most abundant protein found in a SHIV particle, SIV Gag; typically, ~2,000 molecules form the virion core. All virus-only control RMs also had proliferative responses in their CD4^+^ and CD8^+^ T-cell populations. Interestingly, RMs pretreated with the low-dose of HGN194-IgG1 (1 mg/kg) had no proliferative T-cell responses to any of the viral proteins. We conclude that only RMs that encountered the challenge virus after treatment with the high-dose of HGN194-IgG1 developed a high enough IC concentration to trigger priming of these SIV Gag-specific T-cell responses [17]. 

In a subsequent study, we have confirmed the induction of SIV Gag-specific T-cell responses in animals given passive immunization followed by SHIV challenge and complete protection with no evidence of viremia ever [8]. These findings led us to postulate that passive immunization with nmAbs against mucosal SHIV challenges results in IC formation. In turn, these ICs are taken up by mechanisms yet to be determined and initiate virus-specific T-cell adaptive immune responses. 

## 6. Protective Mucosal Ab Responses Induced by Active Immunization

### 6.1. Protection against Intravaginal SHIV Challenge through Vaccination with gp41 Virosomes

The most successful vaccine approach to date to protect RMs against intravaginal SHIV challenge involved a mixture of virosomes displaying HIV-1 gp41 antigens on the particle surfaces [36]. The virosome vaccine platform, generated from inactivated influenza virus and devoid of nucleic acids, consists of noninfectious particles. Virosomes have been used clinically with an excellent safety record [37,38]. Bomsel et al. [36] tested a mixture of two different HIV-1 gp41 virosomes in Chinese-origin RMs. One preparation consisted of virosomes displaying P1, a long peptide that represents the membrane proximal external region (MPER) of HIV-1 gp41. The other virosome prep expressed a truncated form of gp41 lacking the immunodominant loop. One RM group was vaccinated twice by the intramuscular route with the virosome mixture followed by two intranasal boosts; another RM group received four intramuscular immunizations. Control RMs were given empty virosomes. Four weeks after the last boost, the animals were challenged repeatedly by the intravaginal route with SHIV_SF162P3_, a tier 2 virus that was heterologous to the immunogens used. RMs given intranasal boosts were completely protected from persistent systemic infection, although sterilizing immunity was not achieved as some animals exhibited occasionally low-level viremia blips. However, none of these animals seroconverted to SIV Gag. Immune correlate analysis revealed that none of the systemic Ab responses tested, including neutralization and ADCC, correlated with protection. Surprisingly, vaginal fluid containing vaccine-induced IgA blocked transcytosis of cell-free virus in vitro, and vaginal IgG exhibited neutralization and ADCC. Thus, only the *mucosal* IgG and IgA, but not their systemic counterparts, correlated with protection from vaginal R5-SHIV challenge. More recently, the Bomsel group described ADCC activity mediated by IgA in vitro involving Fcα receptor type I (FcαRI; also known as CD89)-expressing cells [39]. These interactions between IgA and CD89^+^ effector cells are important and may explain some efficacy data observed with mucosal IgA.

In an independent study, we have largely confirmed these findings using Indian-origin as opposed to Chinese-origin RMs. We sought replicate the study of Bomsel et al. [36] as closely as possible, including performing the upfront heterologous SHIV_SF162P3_ challenges at two different challenge doses through the intravaginal route. RMs primed twice intramuscularly and boosted twice intranasally showed a high degree of protection (between 78% and 87% depending on the readout) during the first repeated intravaginal SHIV challenge phase, where the challenge virus dose was 70,000 times higher than the average HIV-1 content in the semen of infected men as judged by viral RNA copy numbers [40]. Dictated by the published work of Bomsel et al. [36], the SHIV challenge dose then had to be increased by 50%. During this second challenge phase, when the SHIV challenge dose exceeded the median HIV-1 content found in semen of infected men by >100,000 fold, protection against intravaginal SHIV challenges was lost (Drs. Ruth Ruprecht, Samir Lakhashe, Sylvain Fleury et al., unpublished data). 

In sum, the two independently conducted studies are the only ones that achieved a high degree of protection in two different RM subspecies and warrant optimizing the immunogenicity of this vaccine platform. In the interim, the immunogenicity of the gp41 virosomes has been optimized, and these next-generation virosomes are currently undergoing preclinical testing [41].

### 6.2. Mucosal Vaccinations with Live Attenuated Adenovirus Vectors Encoding SIV Antigens in Rhesus Macaques: Delayed Virus Acquisition Correlated with Rectal Vaccine-induced SIgA Responses after Repeated Intrarectal SIV Challenges 

A study published in 2012 by Xiao et al. [42] tested repeated prime/boost strategies with RM-adapted, live attenuated Adenovirus vectors that encoded SIV *env, rev*, and *gag* via different mucosal routes (sublingual, intranasal/intratracheal, intravaginal, and intrarectal). These immunizations were followed by intramuscular boosting with SIVmac251 gp120. No significant prevention of SIV acquisition was observed. However, anti-SIV Env SIgA titers in rectal secretions significantly correlated with delayed SIV acquisition among vaccinees [42]. 

## 7. Highly Exposed Persistently Seronegative (HEPS) Individuals: The Implied Role of Anti-HIV-1 Mucosal IgA Responses

Several clinical studies have examined the role of mucosal IgA from individuals with a history of frequent exposures to HIV-1 but who remained persistently seronegative despite such risk factors; such persons are identified as Highly Exposed Persistently Seronegative (HEPS). As mentioned earlier, humans have two IgA isotypes that exhibit significant differences in the hinge structure and glycosylation. However, the literature describing human IgA responses is difficult to follow and at times contradictory because IgA isotypes are frequently not defined. In addition, mucosal IgA isolation methods often rely on Jacalin resins, which have a high binding affinity for the O-linked oligosaccharides found in the wide-open, flat hinge region of human IgA1 and not in IgA2. Consequently, Jacalin resin-based IgA isolations predominantly yield IgA1 [43,44,45] (reviewed in [15]). Despite this, the literature often refers to this as “IgA” without further isotype characterization. Such technical issues may explain some of the divergent findings reported for HEPS cohorts regarding mucosal anti-HIV-1 IgA. 

Some groups reported a linkage between anti-HIV-1 IgA responses and resistance to HIV-1 acquisition in sexual partners of HIV-1-infected individuals as well as in sex workers [46,47,48,49,50,51,52]; mucosal IgA was found to be directed against gp41 MPER epitopes [48,50]. Devito et al. [51] reported that mucosal IgA isolated from HEPS was able to neutralize HIV-1 strains belonging to different clades. However, other reports found that HIV-1-specific mucosal Abs, especially IgA, were either undetectable or present only in a small percentage of HEPS subjects [46,47,49,52,53] probably due to differences in mucosal Ab isolation or detection methods. 

A more recent study described the results of a blinded analysis performed on foreskin samples collected during the trial “Male Circumcision for HIV Prevention in Rakai, Uganda” [54]. This study sought to test correlates of HIV-1 acquisition risks in foreskin samples following a case-control study design. Jacalin column chromatography was used to isolate IgA from the foreskin swabs, a method that predominantly yields IgA1 as stated above. The ability of IgA from foreskin samples to neutralize HIV-1 in vitro was associated with an Odds Ratio (OR) of 0.21 for uncircumcised men to become HIV-1 infected among those who were still seronegative at their last visit [55]. The data collected from these men are strikingly similar to those obtained in high-risk female sex workers. For women whose cervical/vaginal secretions had neutralizing IgA, the OR for HIV-1 acquisition was 0.31. A later study involving women enrolled in a Pre-Exposure Prophylaxis (PrEP study) also found anti-HIV-1 neutralizing IgA in the cervical/vaginal secretions [56]: the patient population also involved HEPS individuals. Together, these intriguing data from African men and women give a strong indication that mucosal IgA responses, perhaps IgA1 responses, are linked to lowering the risk of HIV-1 acquisition through sexual exposures. 

## 8. Summary and Conclusions

In this review, we have summarized proof-of-concept data that conclusively demonstrate that Igs of the three classes present in mucosal secretions, IgM, IgG, and dIgA, can act as mucosal defenders and block incoming HIV-1 from establishing persistent systemic infection. Passive immunization studies involving anti-HIV-1 IgG and dIgA nmAbs point to special local interactions of mucosal Igs of different classes among each other—resulting in unexpectedly high degrees of protection. Our working hypothesis is that mucus proteins as well as the epithelial tissue environment play key roles in this protection, the mechanism(s) of which need to be elucidated in future studies. The potent protection provided by the mucosally administered nmAbs give impetus to not only induce mucosal antiviral Ab responses of different classes by active immunization, but also to consider further development of topical nmAb prophylaxis in parallel. Lessons learned from the anti-HIV-1 studies will also be of significance for the development of strategies that seek to prevent mucosal transmission of other pathogens. 

## Figures and Tables

**Figure 1 vaccines-07-00194-f001:**
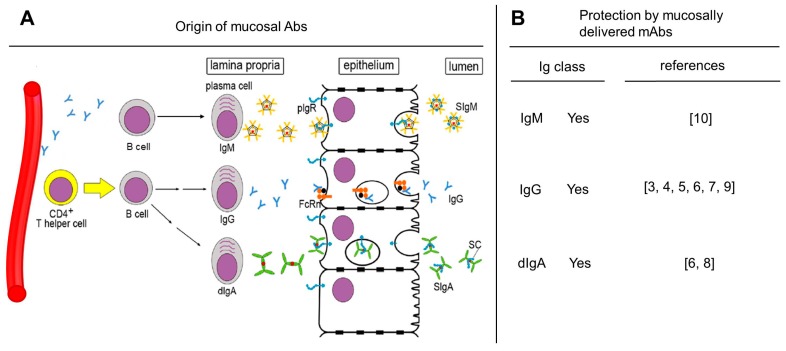
Generation of mucosal immunoglobulins (Igs) and protection of rhesus macaques (RMs) by mucosally delivered anti-HIV-1 envelope recombinant monoclonal antibodies (mAbs). (**A**) Generation and transport of mucosal Igs in a normal host. B cells located in the subepithelial space, the lamina propria, differentiate into plasma cells. IgM-producing plasma cells generate pentameric IgM that contains the Joining (J) chain (red dot) (A, top section). The J chain mediates binding to the polymeric Ig receptor (pIgR; shown in blue). The pIgR-IgM complex shuttles across the epithelial barrier, and at the luminal side, proteolytic cleavage of pIgR leaves behind a membrane-bound stump, whereas the rest of pIgR remains as the secretory component (SC; shown as red dot) with the IgM to form secretory IgM (SIgM). Transepithelial transport of IgG uses a different mechanism (A, middle section). IgG found in mucosal fluids can originate from bone marrow plasma cells; such IgG is taken up by the systemic circulation, and extravasation allows distribution of IgG in tissues. Some of these bone-marrow-derived IgG molecules travel to the lamina propria and bind to the Fc neonatal receptor (FcRn) to be transported across the epithelial barrier. FcRn releases the IgG at the luminal side, generally due to a pH differential. Unlike pIgR, FcRn remains intact and shuttles back and forth between the luminal and the basolateral sides of the epithelium. Mucosal IgG can also be made locally by IgG-secreting plasma cells found in the lamina propria; such locally produced IgG also is shuttled across the epithelium via FcRn. IgG remains unmodified and has no secretory form. Finally, the IgA class of mucosal Igs is produced locally (A, bottom section). With the help of CD4^+^ T-helper cells, B cells in the lamina propria undergo class switching to IgG (middle pathway) or dimeric IgA (dIgA) (bottom pathway). The need for CD4+ T-helper cell participation in class switching is depicted by the tandem black arrows. In contrast, IgM-producing plasma cells do not undergo class switching and continue to function even in the absence of help by CD4^+^ T cells (single black arrow, top pathway). T helper cell-independent mucosal IgA responses have also been described [12,13] (please see text). Lamina propria plasma cells producing dIgA produce molecules that contain two IgA monomers and one J chain (red dot). As described above for pentameric IgM, the J chain in dIgA mediates binding to pIgR, and the dIgA-pIgR complex traverses the epithelial cell to be delivered to the luminal side. As in the case of SIgM, pIgR undergoes proteolytic cleavage to generate secretory IgA (SIgA). In HIV-1 infection, CD4^+^ T-cell function in mucosal compartments is severely compromised, leading to a skewing of ratio of SIgM to IgG and SIgM to SIgA ratios (reviewed in [14]). (**B**) Protection by mucosally delivered anti-HIV-1 Env monoclonal antibodies (mAbs). Passive immunization studies using recombinant mAbs delivered intrarectally (IgM, IgG, dIgA) (or intravaginally only for the IgG form) have demonstrated that all three classes of these mucosal Igs can prevent SHIV transmission. Of note, all mucosal fluids tested contain free secretory component (SC), including mucosal fluids of RMs (Dr. Ruth Ruprecht, unpublished data). To give initial proof-of-concept for protection against mucosal SHIV challenge, the IgM and dIgA classes of recombinant Igs were delivered without SC; all three Ig classes gave clear evidence that mucosal anti-HIV-1 Env mAbs can protect. Most RM challenge studies were conducted with R5-tropic SHIVs. Citations are given in the right column. The schema for passive immunization with recombinant IgM and dIgA nmAbs is shown in Figure 2. This includes showing the presence of free SC in mucosal fluids. Adapted from Kulkarni and Ruprecht, 2017 [14].

**Figure 2 vaccines-07-00194-f002:**
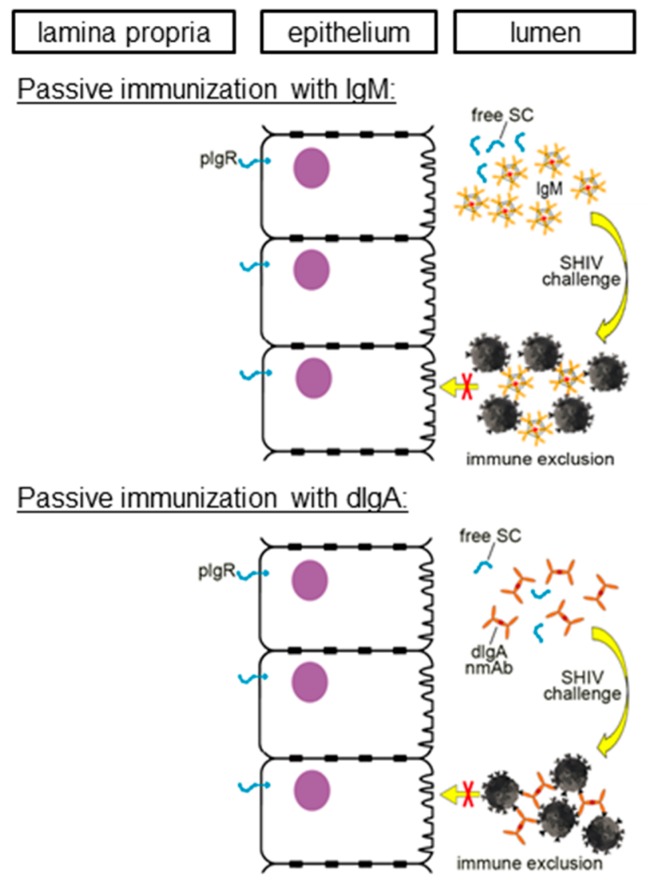
Immune Exclusion: proposed mechanism of protection by mucosally administered recombinant IgM and dIgA mAbs. Recombinantly produced IgM pentamers (top panel) or dIgA1 or dIgA2 neutralizing monoclonal antibodies (nmAbs) (bottom panel) were administered directly into the rectal cavity of rhesus macaques (RMs). Thirty minutes later, the animals were challenged with a single, high-dose of simian–human immunodeficiency virus (SHIV). The first proof-of-concept that IgM can protect against mucosal SHIV challenge was given by our group [10]. Likewise, we also gave the first evidence that dIgA nmAbs can prevent SHIV transmission [6]. In the latter study, dIgA1 provided far superior protection compared to the dIgA2 isotype of the same epitope specificity. To indicate the recombinant nature of the mucosally delivered nmAbs, the IgM and dIgA versions are depicted in ochre and not in the colors shown in Figure 1 where naturally produced mucosal Abs are shown. Adapted from Kulkarni and Ruprecht, 2017 [14].

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
