# Peer review of "Mucosal Antibodies: Defending Epithelial Barriers against HIV-1 Invasion"

_vaccines, 2019, doi:10.3390/vaccines7040194_

Round 1

Reviewer 1 Report

This is a very important review for the HIV vaccine field. As such, the importance of mucosal responses in fighting HIV has been understated, and the importance given to bNAbs has been overstated and narrows perspectives of reaching protection.

This being said, the manuscript could benefit from documenting the contribution of T-independant B-cell responses to mucosal responses (Fig 1A). Indeed, several groups have been working long time on this aspect, which deserves to be aknowledged. 

Also, IgA vs ENV is not always found in mucosal samples of HEPS individuals. In a recent publication, HEPS sex workers were found to present IgG1 vs gp41 in their cervico-vaginal lavages rather than IgA1 or IgA2.

Author Response

1. “This being said, the manuscript could benefit from documenting the contribution of T-independant B- cell responses to mucosal responses (Fig 1A). Indeed, several groups have been working long time on this aspect, which deserves to be acknowledged”.

We thank Reviewer 1 for this suggestion. We have added additional information to the legend of Fig 1A, and we have added several references to document that T helper cell-independent B-cell responses do occur. However, in the case of HIV-1 infection, the marked skewing of the IgM:IgG and IgM:IgA ratios implies that most of the HIV-1-specific mucosal B-cell responses are dependent on T-helper cells. Even early in infection, these T-helper cells are seriously reduced in numbers, which leads to almost undetectable IgA responses in mucosal fluids of some HIV-1-infected individuals. For this reason, we have left the Figure itself unchanged while adding the extra information in the text [page 8, line 166-171] and the legend [Page 22, line 453-454]. It should also be noted that T-independent B-cell mucosal antibody responses seem to predominantly focus on commensal bacteria, rather than viral pathogens such as HIV-1.

2. “Also, IgA vs ENV is not always found in mucosal samples of HEPS In a recent publication, HEPS sex workers were found to present IgG1 vs gp41 in their cervico-vaginal lavages rather than IgA1 or IgA2”.

We agree with Reviewer 1; we actually pointed out that technical problems with IgA isolation and/or determination could skew the findings. We also mentioned that different research groups have reported contradictory results – again because of what we think is mostly technical issues. Please see our comments regarding the use of Jacalin resin to isolate “IgA” [actually, this resin predominantly captures human IgA1 and not very much human IgA2] (page 18, line 384- 390). It is also noteworthy that vaginal fluids generally contain more IgG than IgA in healthy individuals. Depending on the sensitivity of the assays used, the detection of the HIV-1-specific IgA responses could have been below the threshold of detection, whereas IgG was measurable. To emphasize this point, the statement now reads (page 18, line 395-397): “However, other reports found that HIV-1-specific mucosal Abs, especially IgA, were either undetectable or present only in a small percentage of HEPS subjects [45, 46, 48, 51, 52] - probably due to differences in mucosal Ab isolation or detection methods.”

Reviewer 2 Report

Below, please find my review on a review manuscript submitted to Vaccine: special edition, advances in antibody-based HIV-1 vaccine development.

The manuscript is written very clearly. It summarises achievements made by the authors on developing new strategies in preventing HIV infection on mucosal membrane. The authors have summarised several of their studies involving mucosal antibodies.

Minor points:

The text reads very clearly. There are only two typographical errors as below:

page 8 line 68 "does is". please make it clear.

page 11 line 220 "an novel", it should read "a novel".

page 17, line 376, the quotation mark is not closed.

page 18, line 388, whom the unpublished data is owned? please specify the owner of data(i.e. unpublished data, Dr./Prof. …).

Besides above few lines, the only major downside that I could think about the manuscript is lack of literature review on previously published work. For example a direct comparison could be stablished between the reviewed manuscripts and previously published works in line 43, page 3. Report of published works such as:

J Infect Dis. 2016 Aug 15; 214(4): 612–616 / Front Immunol. 2018 Mar 29;9:244 or Nat Med. 2018 May;24(5):610-616

Would be beneficial for the readers.

Author Response

The text reads very clearly. There are only two typographical errors as below:

We thank Reviewer 2 for finding the typographical errors, all of which have been corrected.

page 8 line 68 [sic – presumably the Reviewer meant line 168] "does is". please make it

We apologize for not having deleted “is”. The full sentence should read as follows p8. line XXX: “This results in changes in the IgM:IgG and IgM:IgA ratios in mucosal fluids, given that IgM production does not depend on class switching and the necessary stimulation from functionally intact CD4+ T-helper cells”. [ page 9, line 185]

page 11 line 220 "an novel", it should read "a novel". Corrected [page 11, line 240]

- page 17, line 376, the quotation mark is not closed. Corrected [page 18, line 399]

- page 18, line 388 [sic; presumably 338 is meant], whom the unpublished data is owned? please specify the owner of data(i.e. unpublished data, Dr./Prof. …).”

Page 22, line 465 (Drs. Ruth Ruprecht, unpublished data). We have added the name(s) for all three times where unpublished data were mentioned. Of note, these data were generated by the work of Dr. Ruprecht and her colleagues.

…the only major downside that I could think about the manuscript is lack of literature review on previously published work. For example a direct comparison could be stablished between the reviewed manuscripts and previously published works in line 43, page 3. Report of published works such as:

J Infect Dis. 2016 Aug 15; 214(4): 612–616 / Front Immunol. 2018 Mar 29;9:244 or Nat Med. 2018 May;24(5):610-616

…Would be beneficial for the readers”.

We appreciate the Reviewer’s concern. However, we wish to point out the passive immunizations in nonhuman primate models against mucosal SHIV challenges have been conducted since the late 1990s. The novel aspect of our review is its focus on mucosal antibodies. For that reason, we have refrained from summarizing the experiments in which polyclonal or monoclonal antibodies were administered to experimental animals by the intravenous, intramuscular, subcutaneous or intraperitoneal routes. This is a large body of work, which will distract from the focus on the few key experiments, in which antibodies were unequivocally administered directly into the mucosal lumina. As such, only passive immunizations with mucosally administered antibodies can shed light on the possibility of humoral immune defenses at the epithelial barriers themselves. We noticed that the citations suggested by Reviewer 2 include studies in which the mAbs were delivered intravenously or subcutaneously (Nat Med 2018 May;24(5):610-616, J Infect Dis 2016 Aug 15; 214(4): 612–616), which we could not include. We are, however, citing the work of Duchemin et al. (Front Immunol. 2018 Mar 29;9:244), where ADCC activity mediated by IgA is described in cell culture studies. These interactions between IgA and CD89+ effector cells are important and may explain some efficacy data seen with mucosally delivered IgA [page 16, line 346-349].

Reviewer 3 Report

The authors nicely summarized and covered current data on the role of mucosal antibodies in protection against mucosal SHIV challenges. It would help to clarify two minor points listed below:

1) In Abstract, the 87% dIgA1 and 17% dIgA2 protection was specific to the HGN194 study. So, the antibody HGN194 should be stated. It would also help to specify the challenge SHIV strain.

2) In Section 3, regarding the HGN194 study, the authors commented that the V3 crown-directed mAb HGN194 neutralized all tier 1 strains tested and several tier 2 strains. Did HGN194 neutralize the challenge SHIV strain (again, it would help to specify the challenge SHIV strain)? If it did, was it because the challenge SHIV strain is tier 1?

Author Response

“The authors nicely summarized and covered current data on the role of mucosal antibodies in protection against mucosal SHIV challenges. It would help to clarify two minor points listed below:

In Abstract, the 87% dIgA1 and 17% dIgA2 protection was specific to the HGN194 study. So, the antibody HGN194 should be stated. It would also help to specify the challenge SHIV ”

We thank Reviewer 3 for suggesting this clarification. Given the 200-word limit for the abstract, we had to slightly modify the entire text to accommodate the extra words.

“In Section 3, regarding the HGN194 study, the authors commented that the V3 crown-directed mAb HGN194 neutralized all tier 1 strains tested and several tier 2 strains. Did HGN194 neutralize the challenge SHIV strain (again, it would help to specify the challenge SHIV strain)? If it did, was it because the challenge SHIV strain is tier 1?”

Reviewer 3 is correct. The challenge virus, SHIV-1157ipel-p, is a tier 1 R5 clade C SHIV that was neutralized by HGN194. For easy review, we are including copies of the relevant paper below. We have added an extra explanation on page 10 line 212-213 of the revised manuscript.